

# Impacts of Coal Burning on Ambient PM2.5 Pollution in China

Qiao Ma[1], Siyi Cai[1], Shuxiao Wang[1, 2], Bin Zhao[3], Randall V. Martin[4], Michael Brauer[5], Aaron Cohen[6], Jingkun Jiang[1, 2], Wei Zhou[1], Jiming Hao[1, 2], Joseph Frostad[7], Mohammad H. Forouzanfar[7], Richard T. Burnett[8]

[1]State Key Joint Laboratory of Environment Simulation and Pollution Control, School of Environment, Tsinghua University, Beijing 100084, China
[2]State Environmental Protection Key Laboratory of Sources and Control of Air Pollution Complex, Beijing 100084, China
[3]Joint Institute for Regional Earth System Science and Engineering and Department of Atmospheric and Oceanic Sciences, University of California, Los Angeles, CA 90095, USA
[4]Department of Physics and Atmospheric Science, Dalhousie University, Halifax, Nova Scotia B3H 4R2, Canada
[5]School of Population and Public Health, The University of British Columbia, Vancouver, British Columbia V6T1Z3, Canada
[6]Health Effects Institute, Boston, MA 02110, USA
[7]Institute for Health Metrics and Evaluation, University of Washington, Seattle, WA 98195, USA
[8]Health Canada, Ottawa, ON K1A 0K9, Canada

*Correspondence to*: Shuxiao Wang (shxwang@tsinghua.edu.cn)

**Abstract.** High concentration of fine particles (PM2.5), the primary concern about air quality in China, is believed to closely relate to China's large consumption of coal. In order to quantitatively identify the contributions of coal combustion in different sectors to ambient PM2.5, we developed an emission inventory for the year 2013 using up-to-date information on energy consumption and emission controls, and conducted standard and sensitivity simulations using the chemical transport model GEOS-Chem. According to the simulation, coal combustion contributes 22 μg m$^{-3}$ (40%) to the total PM2.5 concentration at national level (averaged in 74 major cities), and up to 37 μg m$^{-3}$ (50%) in Sichuan Basin. Among major coal-burning sectors, industrial coal burning is the dominant contributor with a national average contribution of 10 μg m$^{-3}$ (17%), followed by coal combustion in power plants and domestic sector. The national average contribution due to coal combustion is estimated to be 18 μg m$^{-3}$ (46%) in summer and 28 μg m$^{-3}$ (35%) in winter. While the contribution of domestic coal burning shows an obvious reduction from winter to summer, contributions of coal combustion in power plants and industrial sector remain at relatively constant levels through out the year.

## 1 Introduction

PM2.5 (particulate matter with aerodynamic diameter less than or equal to 2.5 μm), was considered as the leading air pollutant in most key regions and cities in China, especially in the Beijing-Tianjin-Hebei (BTH) region and the Yangtze River Delta (YRD), according to the air quality status reports released by China's Ministry of Environmental Protection (MEP, 2014a; MEP, 2015). The annual mean PM2.5 concentration in BTH region was 102 μg m$^{-3}$ in 2013 and 93 μg m$^{-3}$ in

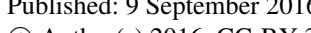


2014, while that in YRD was 67 μg m$^{-3}$ in 2013 and 60 μg m$^{-3}$ in 2014 (MEP, 2014a; MEP, 2015), far beyond even the World Health Organization (WHO) interim target-1 (35 μg m$^{-3}$) for annual mean PM$_{2.5}$ concentration and also the secondary class standard in the China's new National Ambient Air Quality Standard (NAAQS, GB 3095-2012).

The high ambient PM$_{2.5}$ concentration is believed to closely relate to China's large primary energy consumption, especially

coal consumption. According to BP statistical review of world energy (BP, 2015), China has become the largest energy consumer since 2009, of which 2/3 is coal consumption. In the year 2010, coal was responsible for more than 81% of the SO$_2$ emissions, 61% of the NO$_X$ emissions, 40% of the primary PM$_{10}$ emissions, and 34% of the primary PM$_{2.5}$ emissions in China (S. Wang et al., 2014b). As the most abundant and a relatively cheap energy resource, coal is expected to be a dominant energy supply in China in the foreseeable future.

A number of studies have used atmospheric models to study the source contributions of energy use to ambient air pollution in China. Early studies (Wang et al., 2005; Hao et al., 2007) mainly focused on gaseous pollutants, including SO$_2$, NO$_X$, CO and O$_3$. Later on, more studies (Bi et al., 2007; Cheng et al., 2007; Chen et al., 2007; Hao et al., 2007; Wang et al., 2008; Wu et al., 2009) place emphasis on particulate matter, but mainly on PM$_{10}$. Recently, due to the frequent haze episodes characterized by extremely high PM$_{2.5}$ concentration in China, researchers pay more and more attention to PM$_{2.5}$. Among

these studies, most of them took the advantage of 3-D chemical transport models like Community Multi-scale Air Quality Model (CMAQ). H. Zhang et al. (2012) studied source contributions to sulfate and nitrate in PM$_{2.5}$ using the CMAQ model and reported that while power sector is the largest contributor to inorganic components, industry and traffic sector are also important sources. Some recent studies agreed that industrial and domestic sources were the most significant contributors to ambient PM$_{2.5}$ in most areas in China. L. Wang et al. (2014) studied a severe PM$_{2.5}$ pollution episode in Jan. 2013 in North

China for Hebei province using the CMAQ model and concluded that industrial and domestic sources respectively contributed 28% and 27% to local PM$_{2.5}$ concentration. D. Wang et al (2014) conducted simulations with the same model and studied the same pollution episode but for a different city of Xi'an in northwestern China, also reporting that industrial and domestic activities are the two largest sources that accounts for 58% and 16% respectively. L. Zhang et al. (2015) used the GEOS-Chem model and indicated that the residential and industrial sources in North China were respectively responsible

for 49.8% and 26.5% of the PM$_{2.5}$ concentration in Beijing. While most of the studies focused on developed metropolises or heavy pollution episodes, very few studies used atmospheric chemical transport models to study source contributions and its seasonal variation for the whole country throughout a year. In addition, while most researchers studied the total energy consumption in each sector or regarded coal combustion in all sectors as a whole, none of them distinguished coal burning in one sector from another. However, the utilization of coal and the end-of-pipe emission control policies are quite different in

each sector, which leads to different energy efficiency and thus different emissions. Therefore, contributions from coal burning in specific sectors should be identified respectively, which is important for policy making.

In this study, we updated a previously developed emission inventory to the year 2013 using up-to-date information and conducted sensitivity simulations with the chemical transport model GEOS-Chem. In order to obtain a comprehensive understanding of the current contribution from coal combustion to PM$_{2.5}$ concentrations in China, we quantitatively



identified source contributions from coal burning in each sector and its seasonal variation. Section 2 discusses the development of emission inventory for the year 2013; section 3 describes the method of simulation, GEOS-Chem model and its evaluation; section 4 discusses the model results; the last section summarizes the conclusions.

## 2 Emission inventory

Our previous studies have developed the emission inventory of sulfur dioxide ($SO_2$), nitrogen oxide ($NO_X$), $PM_{10}$, $PM_{2.5}$, black carbon (BC), organic carbon (OC), non-methane volatile organic compounds (NMVOC), and ammonia ($NH_3$) for China for the year 2010 using a *technology-based emission factor method* (S. Wang et al., 2014b; Zhao et al., 2013a; Zhao et al., 2013b; Zhao et al., 2013c). The emissions from each sector in each province were calculated from the activity data (energy consumption, industrial products, solvent use, etc.), technology-based emission factors, and penetrations of control

technologies. In this study, we updated the 2010 emission inventory to year 2013 by incorporating the most recent information. The activity data and technology distribution for each sector were updated to 2013 according to the Chinese Statistics (NBS, 2014a; NBS, 2014b; NBS, 2014c) and a wide variety of technology reports (Fu et al., 2015; S. Wang et al., 2014b; CEC, 2011; ERI, 2010; ERI, 2009; THUBERC, 2009). The emission factors used in this inventory were described in Zhao et al. (2013b). The penetrations of removal technologies were updated to 2013 according to governmental bulletins and

the evolution of emission standards (MEP, 2014b).

There are some significant updates for $NH_3$ emissions in this inventory. For agricultural fertilizer application, the emissions of $NH_3$ in the previous study were based on pre-defined emission factors that lacked temporal or spatial details in previous studies. In this inventory, we use an agricultural fertilizer modeling system that couples the regional air quality model CMAQ and an agro-ecosystem model (the Environmental Policy Integrated Climate model, EPIC) to improve the accuracy

of spatial and temporal distribution (Fu et al., 2015). For livestock, the activity data were calculated by the amount of livestock slaughter per year in previous studies. However, the survival periods for each livestock are different and not only one year, thus the amount of slaughter cannot stand for the amount of livestock accurately. In this study, we use the amount of livestock stocks to calculate $NH_3$ emissions and improve the accuracy of the results.

In 2013, the anthropogenic emissions of $SO_2$, NOx, $PM_{10}$, $PM_{2.5}$, BC, OC, NMVOC and $NH_3$ in China were estimated to be

23.2 Mt, 25.6 Mt, 16.5 Mt, 12.2 Mt, 1.96 Mt, 3.42 Mt, 23.3 Mt, and 9.62 Mt, respectively. Table 1 shows emissions by sector and emissions originating from coal combustion, which indicates that in sectors of power plants and domestic fossil fuel combustion, the share of coal-burning emissions are almost over 90%. Coal dominates the emissions in industrial sector as well. In the year of 2013, coal is responsible for 79% of the $SO_2$ emissions, 54% of the $NO_X$ emissions, 40% of the primary $PM_{10}$ emissions, and 35% of the primary $PM_{2.5}$ emissions, 40% of the BC emissions and 17% of the OC emissions.



## 3 Model and simulation

### 3.1 Simulation method

In this study, we conducted one standard simulation and 4 sensitivity simulations for ground level $PM_{2.5}$ using the nested grid capability of GEOS-Chem for East Asia. The simulation scenarios are summarized in Table 2. In the standard

simulation, we use the emissions for the year 2013 that are discussed in Section 2. To select the year of meteorology, we conducted standard simulation using the same emissions and different meteorology from the year 2010 to 2012, as the meteorological fields are not available for the whole year of 2013. We chose the year 2012 as our meteorology year, with which the simulation results best represented the mean $PM_{2.5}$ concentration from 2010 to 2012.

In sensitivity scenarios, we respectively removed emissions from coal combustion in different sectors. In sensitivity scenario

1, we removed emissions from coal burning from all energy sectors (scenario for total coal burning, TC). In sensitivity scenarios 2 to 4, we respectively shut down emissions from coal burning in power plants, industries and domestic sectors (TCP, TCI and TCD). All the meteorology used in the sensitivity simulation was the same as the standard simulation. Three months before each simulation year were used as spin-up. The differences between standard and sensitivity simulations are used to represent the contributions from coal in each sector.

### 3.2 Model description

GEOS-Chem is a global chemical transport model that has been widely applied to study $PM_{2.5}$ over China (e.g. Brauer et al., 2012, 2015; Jiang et al., 2015; Kharol et al., 2013; van Donkelaar et al., 2010, 2015; Y. Wang et al., 2013, 2014; Xu et al. 2015; L. Zhang et al. 2015; Q. Zhang et al., 2015). The model is driven by assimilated meteorological data from the United State National Aeronautics and Space Administration (NASA) Goddard Earth Observing System (GEOS), including winds,

temperature, clouds, precipitation, and other surface properties. GEOS-Chem (version 9-01-03) includes detailed $HO_X$-$NO_X$-VOC-ozone-$BrO_X$ tropospheric chemistry originally described by Bey et al. (2001a) with addition of $BrO_X$ chemistry by Parrella et al. (2012). Aerosol simulation is fully coupled with gas-phase chemistry, including sulfate ($SO_4^{2-}$), Nitrate ($NO_3^-$), and ammonium ($NH_4^+$)(Park et al., 2004; Pye et al., 2009), OC and BC (Park et al., 2003), sea salt (Alexander et al., 2005), and mineral dust (Fairlie et al., 2007). The areasol thermodynamic equilibriums use the ISORROPIA II model (Fountoukis

and Nenes, 2007) to calculate the partitioning of nitric acid and ammonia between gas and aerosol phases. The formation of secondary organic aerosol (SOA) includes the oxidation of isoprene (Henze and Seinfeld, 2006), monoterpenes, aromatics (Henze et al., 2008) and other reactive VOCs (Liao et al., 2007). In addition, we corrected errors in the model representation of too shallow nighttime mixing depth following Walker et al. (2012) and introduced the production mechanism of sulfate on aerosol surface described in Wang et al. (2013). Aerosols interact with gas-phase chemistry in GEOS-Chem through the

effect of aerosol extinction on photolysis rates (Martin et al., 2003) and heterogeneous chemistry (Jacob, 2000).

In this study, we conducted simulations for ground level $PM_{2.5}$ using the nested grid capability of GEOS-Chem for East Asia, which was originally described by Wang et al. (2004) and Chen et al. (2009). The nested domain for East Asia covers area





spanning from 70°E to 150°E, and from 11°S to 55°N, with a horizontal resolution of 0.5 latitudes by 0.667 longitudes. The boundary fields are provided by the global GEOS-Chem simulation with a resolution of 4 latitudes by 5 longitudes and are updated every 3 hours. We assume that the organic mass/organic carbon ratio is 1.8 and relative humidity is 50% for $PM_{2.5}$ in China.

The global simulations use emissions from the Global Emission Inventory Activity (GEIA) inventory (Benkovitz et al., 1996), which is respectively overwritten by the NEI05, EMEP and INTEX-B inventory (Zhang et al., 2009) over the US, Europe, and East Asia. In the nested-grid simulation for East Asia, we use the emissions for the year 2013 as discussed in Section 2 over China, with emissions over the rest of East Asia taken from the INTEX-B emission inventory. In addition, the simulation also includes open fire emissions from GFED3 inventory (Giglio et al., 2010; van der Werf et al., 2010; Mu et al.

2011), lightning $NO_X$ emissions calculated with algorithm of Prince and Rind (1992), volcanic $SO_2$ emissions from AEROCOM data base (http://www-lscedods.cea.fr/aerocom/AEROCOM_HC/) implemented by Fisher et al. (2011).

### 3.3 Model evaluation

GEOS-Chem model is driven by assimilated meteorological data from the NASA GEOS. Y. Wang et al. (2014) has evaluated the important meteorological factors that are relevant to particle formation in the model, including temperature,

relative humidity (RH), wind speed and direction, using observation data from National Meteorological Information Center (NMIC) of China. It reported good spatial and temporal correlations with observed temperature, RH and wind direction. The correlation of wind speed, however, was poorer as the model tends to overestimate in low speed conditions.

In this study, we conducted model evaluation using the surface $PM_{2.5}$ observation network of China National Environmental Monitoring Center (CNEMC, http://106.37.208.233:20035). This monitoring program was initiated in January 2013,

covering 74 major cities in China. Fig. 1 compares simulated annual mean $PM_{2.5}$ concentrations with those observed in 74 major cities in China for the year 2013. As shown in Fig. 1a, the simulated ambient $PM_{2.5}$ concentration has a clear regional distribution with high values in the Sichuan Basin (SCB), North China Plain (NC), and middle Yangtze River area (MYR). The highest concentration occurs in Sichuan Basin with an average value of 73.5 μg m$^{-3}$. Concentrations in the above-mentioned severely polluted regions are generally above 60 μg m$^{-3}$. The observation data are compared with the

concentration of the grids where the city centers are located. The comparison shows that model well reproduces the spatial distribution with a normalized mean bias (NMB) of -16.3%. The correlation efficient for annual mean concentration is 0.68. The slight underestimate mainly appears in heavily polluted area in NC region where observations are largely influenced by local emissions but current simulation cannot capture it with a relatively coarse resolution (H. Zhang et al., 2012). Fig. 2 shows comparisons between simulated and observed seasonal mean concentrations. $PM_{2.5}$ concentration has an obvious

seasonal variation with the highest value in winter and the lowest in summer, which is correctly reproduced by the model. The largest bias occurred in winter with the value of -23.3%. The inconsistency of meteorology field also partly account for the underestimate, as the meteorology condition was more unfavorable in Jan. 2013. Y. Wang et al. (2014) conducted simulations for Jan. in 2012 and 2013 using same emissions, and found the ground $PM_{2.5}$ concentration are 27% higher in





Jan. 2013 than that in 2012. Model performs better in other three seasons, with biases between -13.3% and -10.8%. The seasonal correlation coefficients varied between 0.59 and 0.71.

We also evaluated the monthly variation using averaged monthly mean concentrations in cities in each key region, as analyses and discussions mainly focused on these six areas. The six key regions are shown with frames in **Fig. 1a**, which

includes Northeast China (NEC, 123°E-128°E, 41°N-47°N), North China (NC, 113°E-119°E, 33°N-40°N), Yangtze River Delta (YRD, 119°E-122°E, 29.5°N-32.5°N), Middle Yangtze River (MYR, 111°E-115°E, 27°N-32.5°N), Sichuan Basin (SCB, 103°E-107°E, 28°N-32°N) and Pearl River Delta (PRD, 112°E-114°E, 22°N-24°N). Cities in each region share the similar weather condition, terrain and pollution levels. As shown in Fig. 3, the model generally well reproduces the monthly variation. The NMB ranges from -45% to 1%, and the correlation coefficient varies between 0.7 and 0.94. The model

performance is better in MYR, SCB and PRD than that in NC, NEC and YRD. The large discrepancy is mainly due to the failure to capture the extremely high concentration in wintertime.

The $PM_{2.5}$ composition shows a great diversity across China. Sulfate-nitrate-ammonium (SNA), BC, Organic Matter (OM), and crustal material respectively constituted 7.1% to 57%, 1.3% to 12.8%, 17.7% to 53% and 7.1% to 43% in $PM_{2.5}$ mass in China, and the fractions of SNA in $PM_{2.5}$ (40% - 57%) is much higher in East China (Yang et al., 2011). OM and mineral

dust also play significant roles in $PM_{2.5}$ concentration. $PM_{2.5}$ speciation in China simulated by GEOS-Chem has been evaluated in some previous studies. Wang et al. (2013) reported annual biases of -10%, +31%, and +35% for sulfate, nitrate and ammonia respectively, compared with observations at 22 sites in East Asia. Fu et al. (2012) indicated that annual mean BC and OC concentrations in rural and background sites were underestimated by 56% and 75%. $PM_{2.5}$ speciation is also evaluated in this study using the observed concentration of aerosol compositions from 2006 to 2007 in 16 cities across China

(X. Zhang et al., 2012), which is shown in Fig.4. The model underestimates the annual mean concentration of sulfate, ammonium, BC and OC by 58%, 13%, 34% and 49% respectively, and overestimates nitrate concentration by 2%. The correlation coefficients range between 0.71 and 0.84. However, the $SO_2$ emissions were estimated to be 27826 kt in 2006 and 26455 kt in 2007, and it decreased to 23129 kt in 2013 (S. Wang et al., 2014b). In contrast, the $NO_X$ emissions increased to 25623 kt in 2013 from 20791 kt in 2006 and 22287 kt in 2007 (Zhao et al., 2013c). Considering the evident change of $SO_2$

and $NO_X$ emissions in China from 2006 to 2013, the underestimate for sulfate should be less than 58% and the overestimate for nitrate is higher that 2%.

## 4 Source contributions to ambient $PM_{2.5}$ concentration

### 4.1 Annual mean source contributions

Fig. 5 shows the spatial distribution of annual mean source contributions from coal burning. As shown in Fig. 5a, the

contribution from total coal burning has a similar spatial distribution with the annual mean $PM_{2.5}$ concentration, which indicates the large influence of coal burning on air quality. Table 3 also shows a higher percentage contribution in areas with higher $PM_{2.5}$ concentrations such as NC, MYR and SCB regions. The national average contribution from total coal burning,





which is an average of concentrations in 74 major cities, is up to 22.5 μg m$^{-3}$, which accounts for almost 40% of the total PM$_{2.5}$ concentration. In the six key regions, coal burning contributes 34.5% to 50.2% of the total ambient PM$_{2.5}$ concentration. The largest contribution occurs in SCB, which reaches 36.9 μg m$^{-3}$ on average, due to the dense population, large emissions and unfavorable terrain that tends to trap the emissions and secondary pollutants in this area. The highest

contribution is up to 56.9 μg m$^{-3}$, occurring in the southwest city of Chengdu. Following SCB, coal-burning contributions in MYR and NC are also above the national average, with average values of 30.8 μg m$^{-3}$ (45.1%) and 26 μg m$^{-3}$ (40.5%), respectively. Among the six key regions, coal combustion in PRD shows the smallest contribution of 12.6 μg m$^{-3}$, yet still accounting for 35% of the local PM$_{2.5}$ concentration. In addition to the key regions, coal burning contributes around 25 μg m$^{-3}$ (more than 50%) of the local PM$_{2.5}$ in cities like Baotou and Hohhot in Inner Mongolia, an autonomous region near the

middle north border, as it is one of the largest production areas of coal and a large amount of raw coal is burnt for energy supply. In the northwest city of Urumqi, coal burning is also a large contributor for it accounts for around 40% of the local PM$_{2.5}$ concentration as there are no other large anthropogenic sources of air pollutants there.

Among all the subsectors in coal combustion, industrial coal burning is the most significant contributor, followed by coal burning in power plants and domestic sector, which is shown in Fig. 5b - d and Table 3. The contribution from industrial

coal burning is up to 9.6 μg m$^{-3}$ (17%) on national average (74 major city average), while those from coal burning in power plants and domestic sector are 5.6 μg m$^{-3}$ (9.8%) and 2.2 μg m$^{-3}$ (4%), respectively. The contribution from each sector differs in different regions. Contributions from coal burning in power plants and industry have similar spatial distributions with the annual mean PM$_{2.5}$ concentration. As shown in Fig. 5b, coal burning in power plants has the largest contribution in NC with the highest value of 13.1 μg m$^{-3}$ (15%) and an average of 7.7 μg m$^{-3}$ (12%), due to the large number of power plants in this

area. The smallest contribution occurs in PRD with the value of only 2.7 μg m$^{-3}$ (7.5%). In most key areas in China, coal burning in power sector contributes around 10% of the local PM$_{2.5}$ concentration, which is a relatively minor source compared with industry due to higher energy efficiency and more stringent emission control policies in power sectors. Industrial coal burning, as shown in Fig. 5c, has the largest contribution in SCB, with an average value of 19 μg m$^{-3}$ (25.9%). The largest contribution occurs in the city of Chengdu, which is up to 35.8 μg m$^{-3}$, accounting for around 1/3 of the local

PM$_{2.5}$. NC and MYR are also significantly influenced by industrial coal burning with the contributions of 10.8 μg m$^{-3}$ (16.8%) and 14 μg m$^{-3}$ (20.5%), respectively. In other areas including NEC, YRD and PRD, the average contributions of coal burning in industrial sector are generally less than 10 μg m$^{-3}$, accounting for around 15% of the local PM$_{2.5}$ concentration. As shown in Fig. 5d, domestic coal burning has little contribution to ambient PM$_{2.5}$ in most areas in the six key regions. However, in some individual regions in Guizhou province in Southwest and Inner Mongolia in North China, domestic coal burning

contributes more than 10 μg m$^{-3}$, which accounts for more than 15% in Guizhou and 25% in Inner Mongolia where people tend to burn more raw coal for heating. Besides, the high sulfur content of coal in Guizhou province also accounts for the large contribution.





### 4.2 Seasonal variation of coal contributions

Fig. 6 shows the simulated seasonal mean $PM_{2.5}$ concentration (Fig. 6a and b) and source contributions from coal burning in winter (averaged from December to February) and in summer (averaged from June to August) (Fig. 6c to j), which is also summarized in Table 4 and 5. As shown in Fig. 6a and b, the ambient $PM_{2.5}$ concentration has obviously different distributions in winter and in summer. $PM_{2.5}$ in winter has a similar distribution with the annual mean, but with much higher values. The highest value still occurs in SCB with an average of 118.8 $\mu g\ m^{-3}$ due to the large emission, unfavorable terrain and weather condition in winter. Following SCB, the average concentrations in MYR and NC regions are above 100 $\mu g\ m^{-3}$ and 90 $\mu g\ m^{-3}$, respectively. There are also several populated cities in NEC, where $PM_{2.5}$ are generally above 75 $\mu g\ m^{-3}$ and up to 150 $\mu g\ m^{-3}$. $PM_{2.5}$ in summer has an obviously different distribution from winter with much lower concentrations and more even distribution through out the country due to the stronger vertical mix, more wet deposition and lower emissions. The largest concentration occurs in NC region with 46.9 $\mu g\ m^{-3}$ on average, followed by SCB with an average of 44.1 $\mu g\ m^{-3}$. In addition to the above two regions, $PM_{2.5}$ concentrations in other key regions are generally around or below 35 $\mu g\ m^{-3}$ on average.

In winter, coal burning contributes 28.2 $\mu g\ m^{-3}$ (35.4%) to total $PM_{2.5}$ concentration on the national level. Similar with the annual mean, coal-burning contribution in winter peaks in SCB with an average of 50.3 $\mu g\ m^{-3}$ (42.3%) and reaches the lowest in PRD with 16.1 $\mu g\ m^{-3}$ (29%). Among the coal-burning sectors, the contributions from power plants and industry also have similar spatial pattern with the annual mean distribution. Coal burning in industry, followed by that in power plants, is the largest contributor in both seasons. Domestic coal burning is a significant contributor in winter, due to the large amount of emissions from heating supply. The high $PM_{2.5}$ concentration from domestic sector mainly occurs in some areas in Guizhou Province in southwest and Inner Mongolia in north, where a large amount of raw coal is burnt for heating. The largest contribution reaches as much as 37.6 $\mu g\ m^{-3}$ in Inner Mongolia, which accounts for almost 40% of the local $PM_{2.5}$ concentration.

In summer, the national average contribution from coal burning is estimated to be 17.8 $\mu g\ m^{-3}$ (46.2%), which is less than 2/3 of the contribution in winter, due to the favorable meteorological condition including stronger convection and more frequent wet deposition. Regional contribution ranges from 8.2 $\mu g\ m^{-3}$ in PRD to 26.3 $\mu g\ m^{-3}$ in SCB, which is approximately half of the contributions in winter. The seasonal variation of contributions in inland areas (NEC, MYR, SCB) is more significant than those in coastal areas (NC, YRD, PRD). In coal-burning sectors, the absolute contributions from power plants and industry doesn't show very noticeable reductions in summer compared with those in winter, as emissions from these two sectors are in a relatively constant status throughout the year and the nitrate reduction due to the high temperature in summer is counteracted by the enhancement of the sulfate formation (H. Zhang et al., 2012). In contrast, domestic sector contributes 1 $\mu g\ m^{-3}$ (2.5%) on the national level in summer, which is 3 to 8 times less than that in winter.



### 4.3 Comparisons with other studies

The Natural Resources Defense Council (NRDC) launched the China Coal Consumption Cap Project in Oct. 2013 and released the report of *contribution of coal use to air pollution in China* as part of the study results in October 2014 (NRDC, 2014). This study used the CAMx model with MEIC inventory and meteorology from WRF to simulate coal contributions to

ambient $PM_{2.5}$ in January, February, April and October in the year 2012 in 333 main cities in China. In order to compare with the NRDC study, we extracted the simulated contribution in the 333 main cities during the same periods from our study results. Fig. 7 represents the comparison in each province and shows that our study underestimates the coal contribution by 22% compared to that in the NRDC study. The discrepancy is mainly generated from the different amounts of emissions that are originated from coal in the two studies. According to the report, the NRDC study included both emissions directly from

coal burning and emissions from industries closely related to coal burning. For example, air pollutants from industries like coke, steel, cement and non-ferrous metal are generated from ways: directly from coal combustion and from the technological process. As coal is used as fuel in these industries and is not likely to be substituted for in the near future, the NRDC study includes both the two parts as emissions from *coal use*. In our study, we include only the first part of the emissions as the contribution from coal, which is actually generated from *coal burning*. According to the report by NRDC,

coal combustion is responsible for 79% of the $SO_2$ emissions, 57% of the $NO_X$ emissions and 44% of the primary PM emissions, and the coal-related sources are responsible for 15%, 13% and 23% of the $SO_2$, $NO_X$ and PM emissions. Despite of the difference definition of coal contribution to air pollutant emissions, the NRDC and our study both predicted high contribution to $PM_{2.5}$ concentration from coal, especially in the Municipality of Chongqing and Sichuan province in SCB.

### 4.4 Uncertainty analysis

The uncertainties of the contribution estimates in this study may arise from the uncertainties of the emission inventory, model simulation and non-linearity of the atmospheric chemistry. A Monte Carlo uncertainty analysis was performed on the emission inventory, as described in Zhao et al. (2013c) and S. Wang et al. (2014b). Table 6 shows the uncertainty analysis of the emissions in China. Among all the coal-consuming sectors analyzed in this study, domestic sector is subject to the highest uncertainty, which may lead to more uncertainty in the $PM_{2.5}$ simulation and contribution estimates. Another

important cause of uncertainty is the model simulation of the $PM_{2.5}$ composition. The coal contribution to sulfate is larger than that to nitrate, as the share of coal-burning emissions of $SO_2$ is 79% in this study, 25% higher than that of $NO_X$ emissions. Therefore, the actual coal-burning contribution to $PM_{2.5}$ is very likely to be larger than the estimates in this study, due to the underestimation of sulfate concentration and overestimation of nitrate concentration by the model. In addition, due to the non-linear response of $PM_{2.5}$ concentration to precursor emissions, contributions from coal burning in each sector add

up to less than the contribution from the total coal burning, which means the probable underestimation of the contribution in subsectors. The impact of non-linearity of the atmospheric chemistry on $PM_{2.5}$ concentrations and its composition has been discussed in detail in previous studies (Zhao et al., 2013b; S. Wang et al., 2014a).

## 5 Conclusion

We updated China's emission inventory to the year 2013 using up-to-date information on energy statistics and emission control policies. The anthropogenic emissions of $SO_2$, NOx, $PM_{10}$, $PM_{2.5}$, BC, OC, NMVOC and $NH_3$ in China were estimated to be 23.2 Mt, 25.6 Mt, 16.5 Mt, 12.2 Mt, 1.96 Mt, 3.42 Mt, 23.3 Mt, and 9.62 Mt, respectively. Using the

emission inventory, we conducted standard and sensitivity simulations for major coal-burning sectors to quantitatively identify the source contributions from coal burning using the chemical transport model GEOS-Chem. Results show that coal combustion contributes 22.5 µg m$^{-3}$ (40%) of the total $PM_{2.5}$ concentration on national average (74 major city average). The highest contribution occurs in Sichuan Basin, which reached 36.9 µg m$^{-3}$ and accounts for more than 50% of the local $PM_{2.5}$. Among the subsectors of coal combustion, industrial coal burning is the dominant contributor, with the largest contribution

of 19 µg m$^{-3}$ (26%) in Sichuan Basin and the second largest of 14 µg m$^{-3}$ (20%) in Middle Yangtze River area, which indicates that coal combustion in industry should be prioritized when energy policies and end-of-pipe control strategies are applied, especially in middle-west regions in China, from the perspective of the whole country. Coal combustion in power plants shows the largest contribution in North China with an average of 7.7 µg m$^{-3}$ (12%). Domestic coal burning has the largest contribution in some regions in Guizhou province in Southwest China and Inner Mongolia in North China, where

combustion of raw coal should be substantially reduced especially in winter. An obvious seasonal variation is also predicted. The absolute contributions due to coal combustion are estimated to be 28 µg m$^{-3}$ (35%) in winter and 18 µg m$^{-3}$ (46%) in summer on the national level. The seasonal differences are mainly due to the dramatic change of domestic emissions and more favorable meteorological conditions including stronger convection and wet deposition in summer. While contribution from domestic coal shows a significant reduction from winter to summer, the absolute contributions from coal burning in

power plants and industry remain at relatively steady levels throughout the year.

### Acknowledgement

This work was financially supported by MEP's Special Funds for Research on Public Welfare (201409002), Strategic Priority Research Program of the Chinese Academy of Sciences (XDB05020300), Global Burden of Disease - Major Air Pollution Sources (GBD-MAPS, HEI004GBDTS), and National Natural Science Foundation of China (21521064).

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

| | SO$_2$ | NOx | PM$_{10}$ | PM$_{2.5}$ | BC | OC | NMVOC | NH$_3$ |
|---|---|---|---|---|---|---|---|---|
| Power plants | 6275.4 | 6463.6 | 1034.2 | 612.1 | 8.1 | 14.9 | | |
| Coal[a] | 6209.2 | 6091.2 | 1000.9 | 579.1 | 3.6 | 0.0 | | |
| Industrial combustion | 7226.5 | 4399.8 | 1536.0 | 1030.1 | 142.8 | 41.2 | 133.5 | |
| Coal[b] | 5972.2 | 2969.4 | 1233.9 | 805.6 | 108.4 | 21.4 | 63.7 | |
| Other industrial process | 2061.1 | 2492.7 | 3173.2 | 1982.3 | 561.2 | 429.3 | 6297.4 | 215.0 |
| Coal[c] | 718.8 | 1758.9 | 1521.8 | 782.8 | 220.2 | 179.7 | 1188.8 | |
| Cement | 1704.0 | 2884.8 | 2985.1 | 1866.7 | 11.3 | 33.9 | | |
| Coal[d] | 1270.8 | 2151.4 | 1224.4 | 843.1 | 8.4 | 25.3 | | |
| Steel | 1859.8 | 532.6 | 1388.3 | 1024.2 | 37.7 | 48.2 | | |
| Coal[e] | 1325.1 | 379.5 | 463.0 | 400.4 | 26.9 | 34.4 | | |
| Domestic fossil fuel combustion | 2887.3 | 609.6 | 1320.9 | 974.4 | 448.1 | 348.5 | 4265.6 | 918.6 |
| Coal | 2692.6 | 554.0 | 1220.4 | 893.0 | 413.4 | 317.4 | 848.0 | |
| Domestic biofuel combustion | 72.4 | 477.9 | 2970.8 | 2878.0 | 503.7 | 1582.9 | | |
| On-road transportation | 644.0 | 5138.2 | 121.2 | 114.8 | 52.4 | 33.5 | 2044.2 | |
| Off-road transportation | 329.5 | 2111.6 | 243.6 | 230.8 | 131.5 | 41.5 | 868.8 | |
| Solvent use | | | | | | | 8155.3 | |
| Biomass open burning | 90.2 | 527.1 | 1747.9 | 1441.6 | 57.7 | 576.6 | 1213.8 | |
| Waste disposal | | | | | | | 387.4 | |
| Livestock farming | | | | | | | | 5489.8 |
| Mineral fertilizer application | | | | | | | | 2997.9 |
| **National total emissions** | **23150.2** | **25638.0** | **16521.2** | **12155.1** | **1955.1** | **3423.1** | **23366** | **9621.3** |
| **Emissions from coal combustion** | **18188.7** | **13904.4** | **6664.4** | **4304.0** | **780.9** | **578.2** | **2100.4** | |

[a] Coal here refers to emissions from coal in the corresponding sector in the above row.

[b, c, d, e] In this study industrial coal combustion includes emissions from these four sector.



**Table 2 Summary for simulation scenarios**

|  | Scenarios |  | Description | Meteorology |
|---|---|---|---|---|
| Standard scenario |  | STD | Standard emission for the year 2013 | 2012 |
|  | 1 | TC | Emissions from total coal burning removed | 2012 |
| Sensitivity scenarios | 2 | TCP | Emissions from coal burning in power plants removed | 2012 |
|  | 3 | TCI | Emissions from coal burning in industry removed | 2012 |
|  | 4 | TCD | Emissions from domestic coal burning removed | 2012 |

**Table 3 Annual mean absolute contributions (µg m$^{-3}$) and percentage contributions from coal burning**

|  | Mean PM$_{2.5}$ | Total coal burning contributions | | Contributions from coal burning in | | | | | |
|---|---|---|---|---|---|---|---|---|---|
|  |  |  |  | Power plant | | Industry | | Domestic | |
| National Average* | 56.7 | 22.5 | 39.6% | 5.6 | 9.8% | 9.6 | 17.0% | 2.2 | 4.0% |
| NEC | 34.5 | 13.2 | 38.3% | 3.6 | 10.4% | 5.3 | 15.3% | 1.8 | 5.3% |
| NC | 64.3 | 26.0 | 40.5% | 7.7 | 12.0% | 10.8 | 16.8% | 1.9 | 2.9% |
| YRD | 52.2 | 18.0 | 34.5% | 5.1 | 9.8% | 7.6 | 14.6% | 0.7 | 1.4% |
| MYR | 68.3 | 30.8 | 45.1% | 6.9 | 10.1% | 14.0 | 20.5% | 2.7 | 3.9% |
| SCB | 73.5 | 36.9 | 50.2% | 5.6 | 7.6% | 19.0 | 25.9% | 4.0 | 5.5% |
| PRD | 36.2 | 12.6 | 35.0% | 2.7 | 7.5% | 5.7 | 15.8% | 0.9 | 2.5% |

* The National average is an average of concentrations in 74 grids where major city centers are located.



**Table 4 Seasonal absolute contributions (μg m⁻³) and percentage contributions from coal burning in winter**

| | Mean PM$_{2.5}$ | Total coal burning contributions | | Contributions from coal burning in | | | | | |
| --- | --- | --- | --- | --- | --- | --- | --- | --- | --- |
| | | | | Power plant | | Industry | | Domestic | |
| National Average* | 79.6 | 28.2 | 35.4% | 6.3 | 7.9% | 9.4 | 11.8% | 4.3 | 5.4% |
| NEC | 53.6 | 20.6 | 38.5% | 5.5 | 10.3% | 6.8 | 12.7% | 4.0 | 7.4% |
| NC | 90.0 | 31.8 | 35.3% | 9.2 | 10.2% | 10.6 | 11.8% | 3.1 | 3.4% |
| YRD | 66.2 | 19.5 | 29.5% | 4.8 | 7.2% | 6.7 | 10.1% | 1.2 | 1.7% |
| MYR | 104.9 | 40.2 | 38.3% | 9.3 | 8.9% | 14.0 | 13.4% | 3.8 | 3.6% |
| SCB | 118.8 | 50.3 | 42.3% | 7.4 | 6.3% | 18.9 | 15.9% | 7.3 | 6.2% |
| PRD | 55.4 | 16.1 | 29.0% | 2.2 | 4.0% | 5.4 | 9.8% | 1.8 | 3.2% |

* The National average is an average of concentrations in 74 grids where major city centers are located.

**Table 5 Seasonal absolute contributions (μg m⁻³) and percentage contributions from coal burning in summer**

| | Mean PM$_{2.5}$ | Total coal burning contributions | | Contributions from coal burning in | | | | | |
| --- | --- | --- | --- | --- | --- | --- | --- | --- | --- |
| | | | | Power plant | | Industry | | Domestic | |
| National Average* | 38.4 | 17.8 | 46.2% | 5.2 | 13.4% | 9.0 | 23.4% | 1.0 | 2.5% |
| NEC | 20.3 | 8.9 | 44.1% | 2.7 | 13.3% | 4.8 | 23.4% | 0.5 | 2.5% |
| NC | 46.9 | 21.7 | 46.4% | 7.3 | 15.5% | 10.5 | 22.5% | 1.0 | 2.1% |
| YRD | 34.1 | 14.2 | 41.5% | 4.7 | 13.8% | 6.7 | 19.5% | 0.18 | 0.5% |
| MYR | 36.2 | 20.2 | 56.1% | 5.1 | 14.2% | 11.6 | 32.0% | 1.6 | 4.5% |
| SCB | 44.2 | 26.2 | 59.5% | 4.7 | 10.7% | 16.0 | 38.5% | 1.9 | 4.2% |
| PRD | 20.2 | 8.2 | 40.7% | 2.2 | 10.8% | 4.3 | 21.5% | 0.3 | 1.5% |

5    * The National average is an average of concentrations in 74 grids where major city centers are located.




**Table 6: Results of the uncertainty analysis of the emissions in China.**

|  | NO$_X$ | SO$_2$ | PM$_{2.5}$ | NMVOC |
|---|---|---|---|---|
| Power plants | ±34% | ±30% | ±31% | - |
| Industrial sector | ±41% | ±49% | ±53% | ±63% |
| Residential sector | ±55% | ±51% | ±68% | ±65% |
| Transportation | ±66% | ±48% | ±52% | ±57% |
| Solvent use | - | - | - | ±78% |
| Other sectors[a] | ±177% | ±179% | ±216% | ±184% |
| Total emissions[b] | [-31%,44%] | [-29%,45%] | [-39%,49%] | [-42%,67%] |

[a] Other sectors mainly refer to open biomass burning.

[b] The last line shows the average 90% confidence intervals of the total emissions.

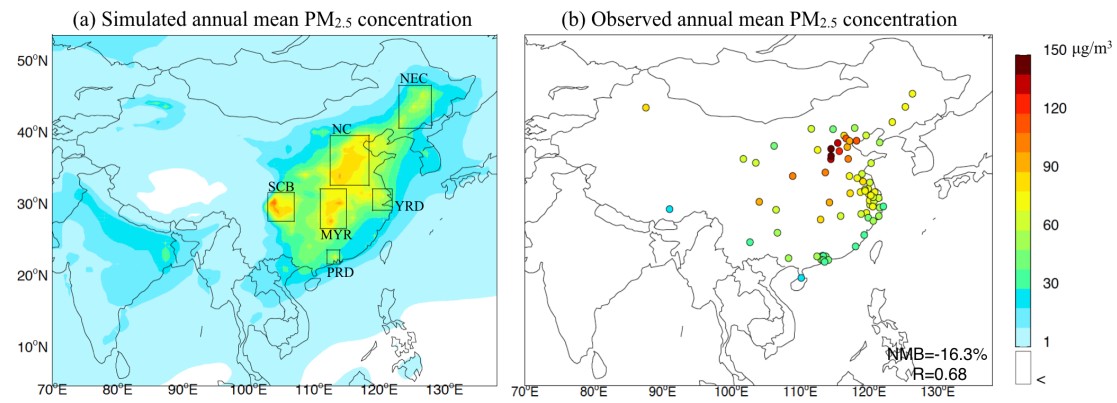

**Figure 1: Simulated and observed annual mean PM$_{2.5}$ concentration in China. The six key regions include the Northeast China (NEC), North China (NC), Yangzte River Delta (YRD), Sichuan Basin (SCB), Middle Yangzte River (MYR), and Pearl River Delta (PRD).**





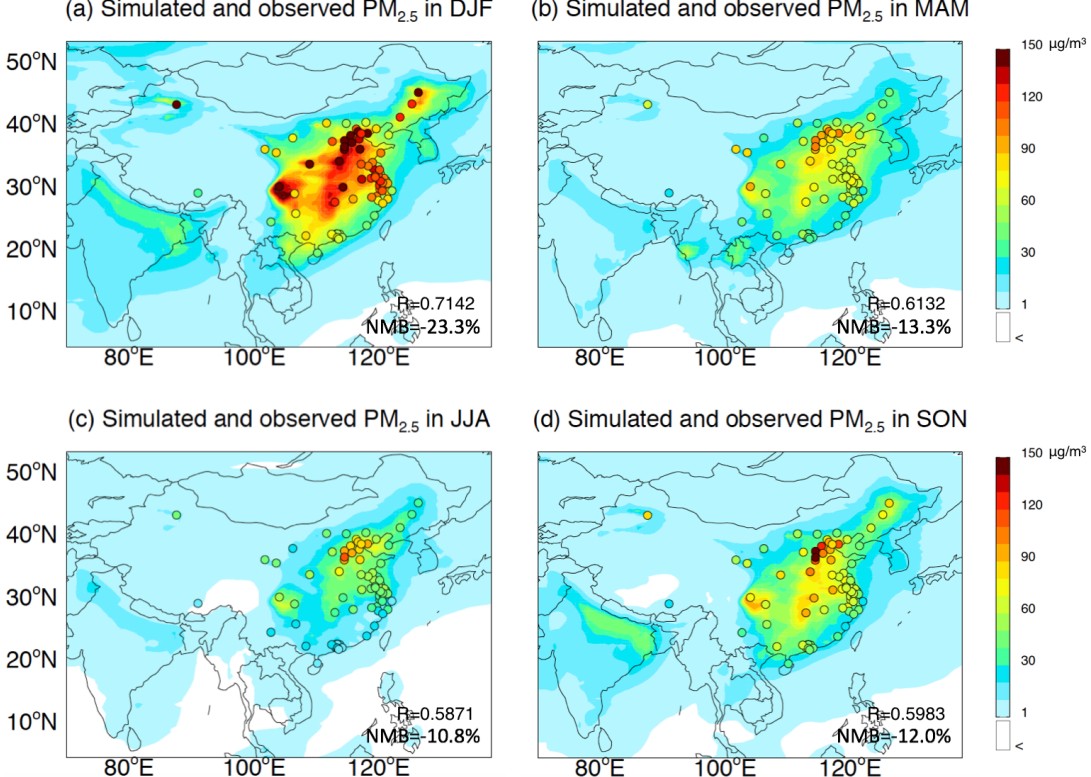

**Figure 2: Simulated and observed sensonal PM₂.₅ concentration in China.**





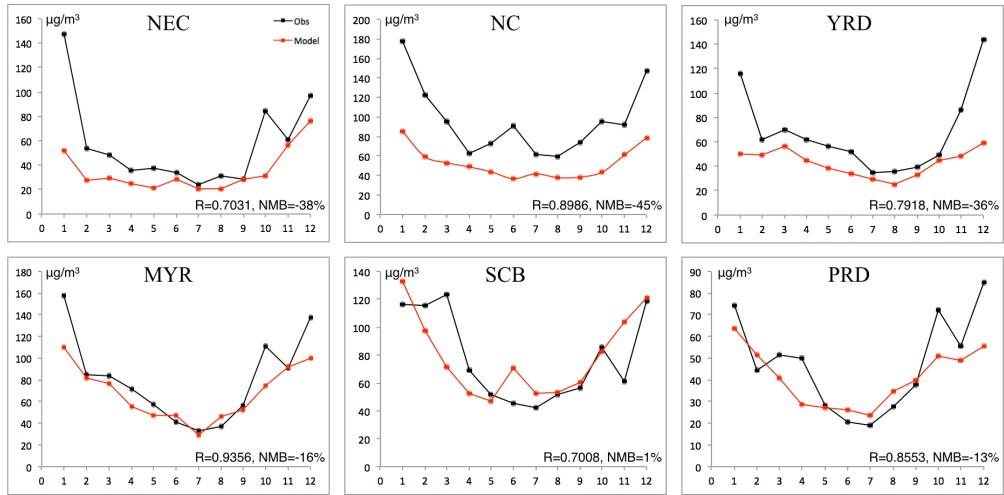

**Figure 3:Monthly mean simulated and observed PM$_{2.5}$ in 6 key regions.**

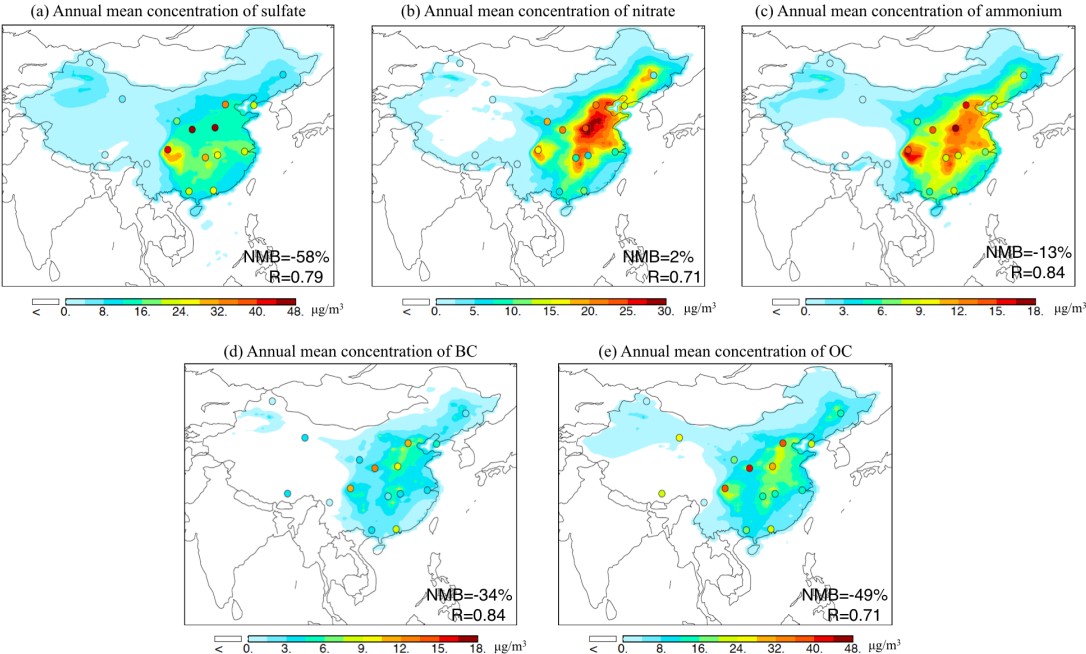

5   **Figure 4:Simulated (2013) and observed (2006-2007) PM$_{2.5}$ speciation in China.**





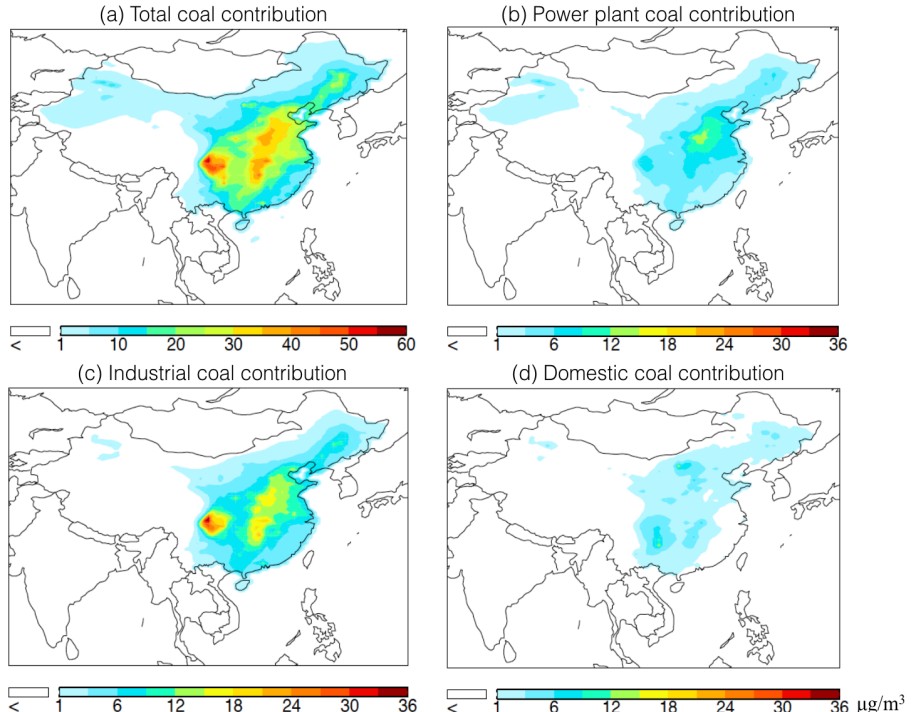

**Figure 5: Annual mean contributions from coal burning.**

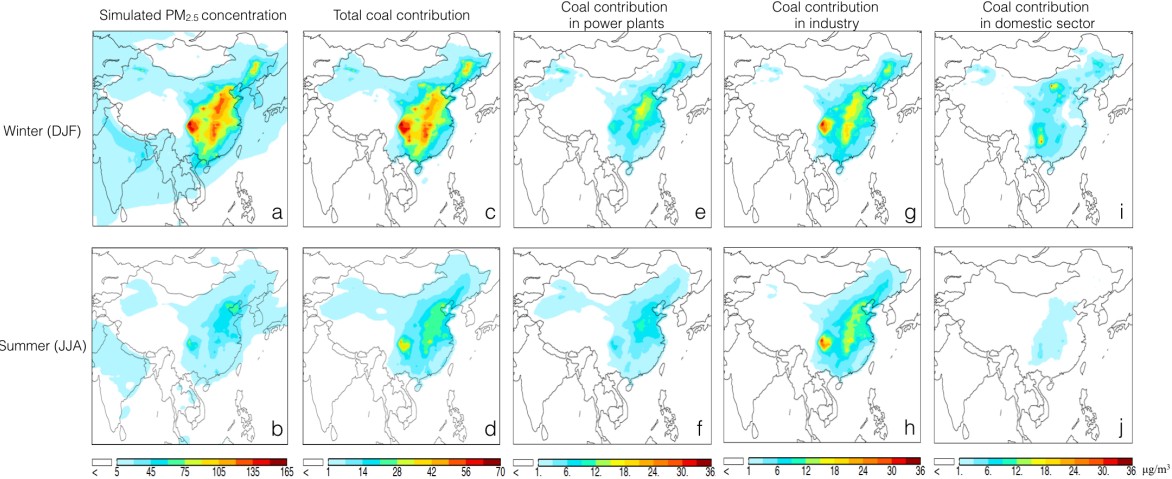

**Figure 6: Seasonal contributions from coal burning in winter and summer.**





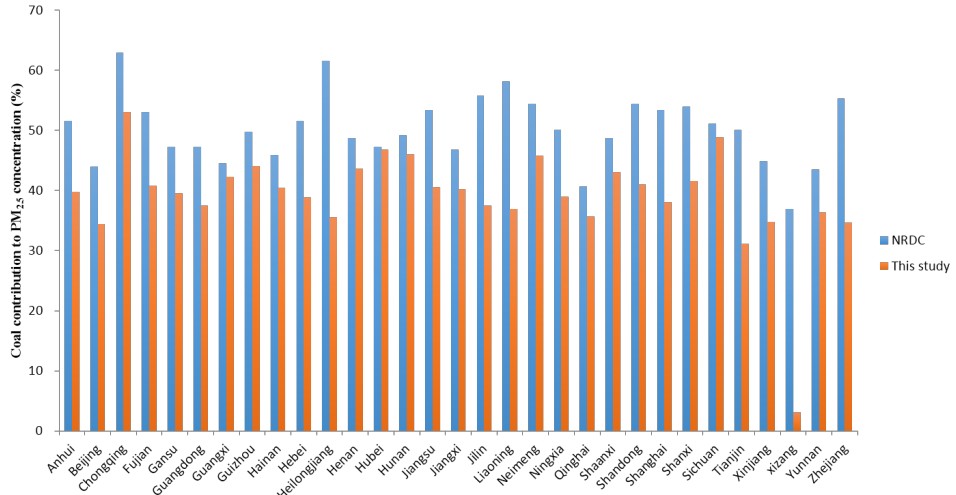

**Figure 7:Comparison of coal contribution to PM$_{2.5}$ concentration between NRDC and this study.**