# Peer review of "Impacts of Coal Burning on Ambient PM2.5 Pollution in China"

_Atmospheric Chemistry and Physics, 2016_

## Referee Comment (RC1) · Anonymous Referee #1 · 16 Oct 2016

Although strict control policies have been conducted, coal burning still dominates China's anthropogenic emissions of air pollutants, and has thereby been a main source of heavy pollution across the country. This manuscript presented a work that integrated the emission inventory development, chemistry transport modeling (CTM), and the sensitivity analysis to find the contribution of coal burning in ambient PM2.5 concentrations. The paper is quite well organized, clearly presented, and easy to follow. I recommend some small revisions or discussions before its publication.

1. Emission inventory: do the authors estimate CO emissions as well or they were obtained from other studies?

2. The model evaluation. The authors used NMB as an indicator, which could potentially be affected by the compensation of overestimation and underestimation of CTM.

I suggest them provide NME for Figs 3 and 4.

3. More discussions should be given in uncertainty analysis. For example, the authors discussed the uncertainty of emission estimations based on Monte-Carlo simulation. However, it was not sufficient for readers to know the impacts of emission inventory estimation on the source apportionment results. More comparisons between various inventory studies are encouraged here to indicate the potential uncertainty of source apportionment from emission side. Moreover, there are some studies using the methods other than Brute-force to reduce the impacts of non-linear response of PM2.5 concentrations to precursor emissions, and they should be included in the part.

4. In general the language is clear, however there are some grammar errors which need to be carefully revised before publication.

---

## Referee Comment (RC2) · Y Balkanski (Referee) · 2 Dec 2016

The authors use an emission inventory that separates the contribution of coal burning to PM2.5 emissions by different activity sectors for the year 2013. This is an update of an existing emission inventory for 2010. THe model uses the nested capability of GEOS-Chem and compares results for surface PM2.5 concentrations to the mesaurements of the China National Environmental Monitoring Center for that same year.

The description of the simulation is too concise for the reader to understand from the information provided if the contributions to PM2.5 from sources outside the nested domain are accounted for or not. If these contributions are accounted for, a paragraph should discuss the importance of these contributions and a Figure should show the relative importance of the sources within the domain and contrast them with outside

sources and their contribution.

The aerosol composition used page 6 has been gathered for measurements taken from 2006 to 2007 in cities across China. Your study centers on the year 2013. How did you connect this composition for 2006-2007 to the year 2013?

Concerning Figure 2, you present the maps of surface PM2.5 for four seasons and simply give the normalized mean biais and the correlation coefficient. I would like to see with Figure 2 the correlation plots so that the reader can have a better view of how the predicted PM2.5 concentations agree/disagree with the measured ones.

Finally the syntax for paragraphs 4.1 through 4.4 should be improved before the manuscript is considered for publication in ACP.

thank you,

YVes Balkanski

---

## Author Comment (AC1) · 12 Jan 2017

1. Emission inventory: do the authors estimate CO emissions as well or they were obtained from other studies?

Response: In this study, we didn't develop emission inventory for CO. The CO emission we used in this study is from EDGAR v3 in global simulations, which is overwritten by INTEX-B (http://mic.greenresource.cn/intex-b2006) in the nested domain of East Aisa.

2. The model evaluation. The authors used NMB as an indicator, which could potentially be affected by the compensation of overestimation and underestimation of CTM. I suggest them provide NME for Figs 3 and 4.

Response: As suggested, we calculated the NME for Fig. 3 and 4 in manuscript. For

[Figure]

Fig. 3, the NME of simulated PM2.5 concentrations in NEC, NC and YRD regions are estimated to be 38%, 45% and 36%, which is the same as the value of NMB, as the model underestimated the PM2.5 concentration throughout the year. In MYR, SCB and PRD regions, the NME are estimated to be 18%, 21%, 22%, which are higher than the estimated NMB, especially in SCB. Overall, the model can reproduce the monthly variation of ambient PM2.5 concentration in these key regions. For Fig. 4 in manuscript, the NME of sulfate, nitrate, ammonium, BC and OC are estimated to be 58%, 41%, 28%, 44% and 50%. The NME of nitrate and ammonia show large difference with NMB. The difference mainly arise from the discrepancies between simulation and observation in NC and MYR. In addition, we add the comparison of simulated PM2.5 composition with observation data averaged during 2012-2013 (X. Zhang et al., 2015), which is shown in Fig. 1 in this reply. The information of each site is described in detail in Zhang et al. (2012). The sulfate is underestimated by 40.5%, which mainly occurs in the two cities of Zhengzhou and Xi'an, two orange spots in central and north China, as these two sites are located in urban area. Nitrate and ammonia are overestimated by around 20%, which is a common issue in most CTMs. OC is underestimated by 28.9% due to the incomplete mechanism of SOA simulation. The NME is calculated between 30% and 41%. Generally the model can reproduced the special distribution of PM2.5 speciation. We add the text and figures in the manuscript as suggested.

3. More discussions should be given in uncertainty analysis. For example, the authors discussed the uncertainty of emission estimations based on Monte-Carlo simulation. However, it was not sufficient for readers to know the impacts of emission inventory estimation on the source apportionment results. More comparisons between various inventory studies are encouraged here to indicate the potential uncertainty of source apportionment from emission side. Moreover, there are some studies using the methods other than Brute-force to reduce the impacts of non-linear response of PM2.5 concentrations to precursor emissions, and they should be included in the part.

Response: We looked into recent studies on major pollutant emissions in China and

summarized them in Table 1 in this reply. Emissions from Liu et al. (2016), Xia et al. (2016) and Wu et al. (2016) are also estimated using bottom-up method, while those from Zhao et al. (2014) are projected emissions for 2015 based on the year of 2010. We can see that the results of this study fall into the range of previous studies except for MEP (2014) which is at low end. One major reason for low NOx emission from MEP(2014) is that it does not include the emissions from non-road vehicles.

Regarding to the non-linearity of atmospheric chemistry, there are some studies using different methods to study the source apportionment of ambient PM2.5. As this study only focuses on coal-burning emissions in each sector, the results are not directly comparable to most similar studies except for results for power sector, as coal combustion dominates the emissions in power plants. Zhao et al. (2015) used the extended response surface modeling (ERSM) technique to access the non-linear response of fine particles to precursor emissions in each sector in PRD region, reporting that local PM2.5 concentration decreased less than 3% (7.2% in our study) in January and around 12% in august (13.8% in our study) when 90% of emissions in power plants are reduced. Our results include the trans-boundary contributions as we shut off emissions across the country in the sensitivity simulation, which is one of the reasons causing the discrepancies. L. Zhang et al. (2015) took the advantage of the adjoint capability of GEOS-Chem, reporting that power plants contributed 6% to PM2.5 concentration in Beijing, which is consistent with our study (6.9%). We also add the above text in the manuscript as suggested.

4. In general the language is clear, however there are some grammar errors which need to be carefully revised before publication.

Response: We proofread the manuscript and revised grammar errors in the text carefully, as suggested.

References

Liu F, Zhang Q, Zheng B, et al. Recent reduction in NO x emissions over China: synthesis of satellite observations and emission inventories[J]. Environmental Research Letters, 2016, 11(11): 114002.

Ministry of Environmental Protection of China (MEP): 2013 Report on the State of Environment in China, http://www.mep.gov.cn/gkml/hbb/qt/201407/t20140707_278320.htm (last access: 16 May 2016), 2014 (in Chinese).

Wu R, Bo Y, Li J, et al. Method to establish the emission inventory of anthropogenic volatile organic compounds in China and its application in the period 2008–2012[J]. Atmospheric Environment, 2016, 127: 244-254.

Xia Y, Zhao Y, Nielsen C P. Benefits of China's efforts in gaseous pollutant control indicated by the bottom-up emissions and satellite observations 2000–2014[J]. Atmospheric Environment, 2016, 136: 43-53.

Zhang L, Liu L, Zhao Y, et al. Source attribution of particulate matter pollution over North China with the adjoint method[J]. Environmental Research Letters, 2015, 10(8): 084011.

Zhang X Y, Wang Y Q, Niu T, et al. Atmospheric aerosol compositions in China: spatial/temporal variability, chemical signature, regional haze distribution and comparisons with global aerosols[J]. Atmospheric Chemistry and Physics, 2012, 12(2): 779-799.

Zhang X Y, Wang J Z, Wang Y Q, et al. Changes in chemical components of aerosol particles in different haze regions in China from 2006 to 2013 and contribution of meteorological factors[J]. Atmospheric Chemistry and Physics, 2015, 15(22): 12935-12952.

Zhao B, Wang S X, Xing J, et al. Assessing the nonlinear response of fine particles to precursor emissions: development and application of an extended response surface modeling technique v1. 0[J]. Geoscientific Model Development, 2015, 8(1): 115-128.

Zhao Y, Zhang J, Nielsen C P. The effects of energy paths and emission controls and standards on future trends in China's emissions of primary air pollutants[J]. Atmospheric Chemistry and Physics, 2014, 14(17): 8849-8868.

[Figure]

[Figure]

Figure 1 Comparisons of simulated PM$_{2.5}$ composition with observation averaged during 2012-2013

**Fig. 1.**

Table 1 Comparisons with other studies on recent air pollutant emissions in China (kt)

|  | SO$_2$ | NO$_X$ | PM$_{10}$ | PM$_{2.5}$ | VOCs |
|---|---|---|---|---|---|
| This study | 23150 | 25638 | 16521 | 12155 | 23366 |
| MEP, 2014 | 20439 | 22273 | - | - | - |
| Liu et al., 2016 | - | 28300 | - | - | - |
| Xia et al., 2016 | 23014-26884 | 28002-28817 | - | - | - |
| Wu et al., 2016 (2012)[*] | - | - | - | - | 29850 |
| Zhao et al., 2014 (2015)[*] | 26792 | 27511 | 15599 | 11419 | - |

*The year of emission are marked in brackets when it is different from the year of emission (2013) in our study.

**Fig. 2.**

---

## Author Comment (AC2) · 12 Jan 2017

1. The description of the simulation is too concise for the reader to understand from the information provided if the contributions to PM2.5 from sources outside the nested domain are accounted for or not. If these contributions are accounted for, a paragraph should discuss the importance of these contributions and a Figure should show the relative importance of the sources within the domain and contrast them with outside

Response: In the nested simulation for East Asia, the contribution from outside the nested domain are accounted for. In order to quantify this contribution, we conducted another sensitivity simulation with all sources outside the domain shut off. The standard and sensitivity simulation results are shown in Fig.1 (a) and (b) in this reply, and the difference between them is analyzed as the contribution from outside the domain,

which is shown in Fig.1 (c). The maximum contribution from outside is up to 13.8 $\mu$g/m3, which mainly occurs in the west and northwest boundaries. The average contributions is 1.57 $\mu$g/m3 in the simulation domain of East Asia. Within the boundary of China, the largest contribution occurs in the Northeast, which is 7.35 $\mu$g/m3. The average contribution from outside the nested domain is only 0.3 $\mu$g/m3 within China We also add the above text and figures in the manuscript as suggested.

2. The aerosol composition used page 6 has been gathered for measurements taken from 2006-2007 in cities across China. Your study centers on the year 2013. How did you connect this composition for 2006-2007 to the year 2013?

Response: We didn't adjust the observation during 2006-2007 to connect to our simulation, but took into account the differences of emissions in 2006, 2007 and 2013, when we interpret the evaluation results. To better resolve this issue, we add the evaluation using the observation data from X. Zhang et al. (2015), which is shown in Figure 2 in this reply. The observed concentration is the average from 2012 to 2013. The information of each site is described in detail in Zhang et al. (2012). The underestimate of sulfate mainly occurs in the two cities of Zhengzhou and Xi'an, two orange spots in central and north China, as these two sites are located in urban area. Nitrate and ammonia are overestimated by around 20%, which is a common issue in most CTMs. OC is underestimated by 28.9% due to the incomplete mechanism of SOA simulation. The NME is calculated between 30% and 41%. Generally the model can reproduced the special distribution of PM2.5 speciation.

3. Concerning Figure 2, you present the maps of surface PM2.5 for four seasons and simply give the normalized mean bias and the correlation coefficient. I would like to see with Figure 2 the correlation plots so that the reader can have a better view of how the predicted PM2.5 concentrations agree/disagree with the measured ones.

Response: We made the correlation plot for each season, as shown in Fig. 3 in this reply. The PM2.5 concentration is more spread out in coordinates in winter as it varies

substantially across China, which has a larger correlation coefficient of 0.71. In other seasons, the correlation coefficients are around 0.6. We also add the above text and figures in the manuscript as suggested.

4. Finally the syntax for paragraphs 4.1 through 4.4 should be improved before the manuscript is considered for publication in ACP

Response: We re-plot the figures from 4.1 to 4.4 and improved the syntax in the manuscript as suggested.

References

Zhang X Y, Wang Y Q, Niu T, et al. Atmospheric aerosol compositions in China: spatial/temporal variability, chemical signature, regional haze distribution and comparisons with global aerosols[J]. Atmospheric Chemistry and Physics, 2012, 12(2): 779-799.

Zhang X Y, Wang J Z, Wang Y Q, et al. Changes in chemical components of aerosol particles in different haze regions in China from 2006 to 2013 and contribution of meteorological factors[J]. Atmospheric Chemistry and Physics, 2015, 15(22): 12935-12952.

[Figure]

(a) Standard simulation for base year    (b) Simulation with zero boundary fields    (c) Contribution from outside the nested domain

Figure 1 Contributions from outside the nested domain

**Fig. 1.**

[Figure]

Figure 2 Comparisons of simulated PM₂.₅ composition with observation averaged during 2012-2013

**Fig. 2.**

[Figure]

Figure 3 Correlation maps for each season

**Fig. 3.**

---

## Author Response (AR2)

**Part 1 Point-to-point responses to the reviewers**

We appreciate the reviewer's valuable comments and point-to-point responses are given below. The original comments are in black, while our responses are in blue.

**5 RC1**

1. Emission inventory: do the authors estimate CO emissions as well or they were obtained from other studies?

- 10 Response: In this study, we didn't develop emission inventory for CO. The CO emission we used in this study is from EDGAR v3 in global simulations, which is overwritten by INTEX-B (http://mic.greenresource.cn/intex-b2006) in the nested domain of East Aisa. We add this information in 3.2 Model description in the manuscript.
- 15 2. The model evaluation. The authors used NMB as an indicator, which could potentially be affected by the compensation of overestimation and underestimation of CTM. I suggest them provide NME for Figs 3 and 4.

Response: As suggested, we calculated the NME for Fig. 3 and 4 in the original manuscript. For Fig. 3, the NME of simulated  $PM_{2.5}$  concentrations in NEC, NC and YRD regions are estimated to be 38%, 45% and 36%, which

20 is the same as the value of NMB, as the model underestimated the PM2.5 concentration throughout the year. In MYR, SCB and PRD regions, the NME are estimated to be 18%, 21%, 22%, which are higher than the estimated NMB, especially in SCB. Overall, the model can reproduce the monthly variation of ambient PM2.5 concentration in these key regions.

For Fig. 4, the NME of sulfate, nitrate, ammonium, BC and OC are estimated to be 58%, 41%, 28%, 44% and

- 50%. The NME of nitrate and ammonia show large difference with NMB. The difference mainly arise from the discrepancies between simulation and observation in NC and MYR.
  In addition, we add the comparison of simulated PM2.5 speciation with observation data averaged during 2012-2013 (X. Zhang et al., 2015), which is shown in Figure R1. The information of each site is described in detail in Zhang et al. (2012). The sulfate is underestimated by 40.5%, which mainly occurs in the two cities of Zhengzhou
- 30 and Xi'an, two orange spots in central and north China, as these two sites are located in urban area. Nitrate and ammonia are overestimated by around 20%, which is a common issue in most CTMs. OC is underestimated by

28.9% due to the incomplete mechanism of SOA simulation. The NME is calculated between 30% and 41%. Generally the model can reproduced the special distribution of  $PM_{2.5}$  speciation. We also add the text and figures in 3.3 Model evaluation in the manuscript as suggested.

Figure R1 Comparison of simulated PM2.5 composition with observation

3. More discussions should be given in uncertainty analysis. For example, the authors discussed the uncertainty of emission estimations based on Monte-Carlo simulation. However, it was not sufficient for readers to know the impacts of emission inventory estimation on the source apportionment results. More comparisons between various inventory studies are encouraged here to indicate the potential uncertainty of source apportionment from emission side. Moreover, there are some studies using the methods other than Brute-force to reduce the impacts of non-linear response of PM2.5 concentrations to precursor emissions, and they should be included in the part.

15

5

Response: We looked into recent studies on major pollutant emissions in China and summarized them in Table R1. Emissions from Liu et al. (2016), Xia et al. (2016) and Wu et al. (2016) are also estimated using bottom-up method, while those from Zhao et al. (2014) are projected emissions for 2015 based on the year of 2010. We can

see that the results of this study fall into the range of previous studies except for MEP (2014) which is at low end. One major reason for low NOx emission from MEP (2014a) is that it does not include the emissions from non-road vehicles.

|                           | 50                      | NO          | DM        | DM                 | VOC-  |
|---------------------------|-------------------------|-------------|-----------|--------------------|-------|
|                           | S O 2 | NOX         | $PM_{10}$ | PIM 2.5 | vocs  |
| This study                | 23150                   | 25638       | 16521     | 12155              | 23366 |
| MEP, 2014                 | 20439                   | 22273       | -         | -                  | -     |
| Liu et al., 2016          | -                       | 28300       | -         | -                  | -     |
| Xia et al., 2016          | 23014-26884             | 28002-28817 | -         | -                  | -     |
| Wu et al., 2016 (2012)*   | -                       | -           | -         | -                  | 29850 |
| Zhao et al., 2014 (2015)* | 26792                   | 27511       | 15599     | 11419              | -     |

**5 Table R1 Comparisons with other studies on recent air pollutant emissions in China (kt)**

\* The year of emission are marked in brackets when it is different from the year of emission (2013) in our study.

Regarding to the non-linearity of atmospheric chemistry, there are some studies using different methods to study the source apportionment of ambient  $PM_{2.5}$ . As this study only focuses on coal-burning emissions in each sector,

- 10 the results are not directly comparable to most similar studies except for results for power sector, as coal combustion dominates the emissions in power plant. Zhao et al. (2015) used the extended response surface modeling (ERSM) technique to access the non-linear response of fine particles to precursor emissions in each sector in PRD region, reporting that local PM2.5 concentration decreased less than 3% (7.2% in our study) in January and around 12% in august (13.8% in our study) when 90% of emissions in power plants are reduced. Our
- 15 results include the trans-boundary contributions as we shut off emissions across the country in the sensitivity simulation, which is one of the reasons causing the discrepancies. L. Zhang et al. (2015) took the advantage of the adjoint capability of GEOS-Chem, reporting that power plants contributed 6% to PM2.5 concentration in Beijing, which is consistent with our study (6.9%).

We also add the above text in the manuscript as suggested.

20

4. In general the language is clear, however there are some grammar errors which need to be carefully revised before publication.

Response: We proofread the manuscript and revised grammar errors in the text carefully, as suggested.

25

**RC2**

5

1. The description of the simulation is too concise for the reader to understand from the information provided if the contributions to  $PM_{2.5}$  from sources outside the nested domain are accounted for or not. If these contributions are accounted for, a paragraph should discuss the importance of these contributions and a Figure should show the relative importance of the sources within the domain and contrast them with outside

Response: In the nested simulation for East Asia, the contribution from outside the nested domain are accounted for. In order to quantify this contribution, we conducted another sensitivity simulation with all sources outside the domain shut off. The standard and sensitivity simulation results are shown in Fig.R2 (a) and (b), and the difference between them is analyzed as the contribution from outside the domain, which is shown in Fig.2R (c). The maximum contribution from outside is up to 13.8 μg/m3, which mainly occurs in the west and northwest boundaries. The average contributions is 1.57 μg/m3 in the simulation domain of East Asia. Within the boundary of China, the largest contribution occurs in the Northeast, which is 7.35 μg/m3. The average contribution from 15 outside the nested domain is only 0.3 μg/m3 within China

We also add the above text and figures in 4.1 in the manuscript as suggested.